# Mining Social Media Data to Capture Urban Park Visitors' Perception of Cultural Ecosystem Services and Landscape Factors

Yaxin Chen [1], Chuanchun Hong [1], Yifan Yang [2], Jiaxin Li [1], Yu Wang [1], Tianyu Zheng [1], Yinke Zhang [3,*] and Feng Shao [1,*]

1. School of Landscape Architecture, Zhejiang Agriculture and Forestry University, Hangzhou 311300, China; chenyaxin@stu.zafu.edu.cn (Y.C.); hongchuanchun@stu.zafu.edu.cn (C.H.); lijiaxin@stu.zafu.edu.cn (J.L.); wangyu8@stu.zafu.edu.cn (Y.W.); zhengtianyu@stu.zafu.edu.cn (T.Z.)
2. Nature Conservation (National Park) Division, East China Academy of Inventory and Planning of National Forestry and Grassland Administration, Hangzhou 310019, China; ivana1804@stu.zafu.edu.cn
3. Hangzhou Botanical Garden, Hangzhou 310012, China
* Correspondence: zykedu@outlook.com (Y.Z.); shaofeng@zafu.edu.cn (F.S.); Tel.: +86-1585-815-8022 (Y.Z.); +86-1345-682-9121 (F.S.)

**Abstract:** Urban parks not only enhance urban ecology but also play a crucial role in providing cultural ecosystem services (CESs) for the well-being of urban residents. Both artificial and natural landscape factors within parks contribute significantly to the supply of cultural ecosystem services. To explore public perceptions of landscape factors and CESs, this study focused on 25 urban parks in Hangzhou. Social media data uploaded by park visitors from 2018 to 2023 were collected to establish a corresponding CES indicator framework. Combining computer vision with text mining, we assessed the preferences and correlations between visitor-perceived CESs and park landscape factors. The results indicated that the majority of park visitors perceive CESs (80.00%) with overall satisfaction higher than importance. Among them, aesthetic experiences and recreation showed both high satisfaction and importance. In shared social media photos, arbors (19.01%), herbaceous flowers (8.99%), and groves (8.22%) were frequently presented as landscape factors. The study revealed close correlations between user gender, landscape factors, and perceived CES categories, with females contributing more to the perception of both. There were internal correlations within CES categories, with spiritual services, aesthetic experiences, and recreation showing the most significant associations. Different landscape factors impacted CES categories to varying degrees, and biological landscapes formed by plant and animal factors were considered to provide more CESs. These findings are significant for enhancing the quality of ecological services and biodiversity in parks.

**Keywords:** urban parks; cultural ecosystem services; landscape factors; public perception; social media data; machine learning

## 1. Introduction

The development of urbanization constrains the natural environment in urban areas. Urban parks have become increasingly important as spaces for urban residents to connect with nature and engage in social interactions [1]. Throughout the lengthy process of urban development, urban parks have accumulated rich and irreplaceable natural and cultural heritage, holding high potential value in aesthetics, ecology, and archaeology [2]. The accumulation of these resources makes urban parks the primary source of urban ecosystem services [3,4]. The Millennium Ecosystem Assessment defined ecosystem services as the benefits people obtained from ecosystems, categorizing them into four main types: supporting, regulating, provisioning, and cultural [5]. As a crucial component of ecosystem services, cultural ecosystem services (CESs) refer to the non-material benefits people derive from ecosystems through spiritual fulfillment, cognitive development, reflection, and aesthetic experiences [5]. They holds enormous potential for improving biodiversity,

human mental and physical well-being, and more [6]. Urban park CESs directly reflect the interaction between humans and ecosystems [7], constituting the subjective perception process of individuals toward objective ecological systems. Effectively understanding residents' perceptions regarding CESs provided by green and blue areas within urban parks is important. It contributes to better providing residents with the specific CESs they need and strengthening residents' commitment to environmental conservation [6,8]. However, due to the subjective and intangible nature of CESs, traditional methods face challenges in objectively assessing and quantitatively analyzing them [9,10]. Therefore, identifying and evaluating CESs pose specific challenges [10,11].

Currently, CES evaluation relies on monetization and non-monetization methods [12]. Recently, more researchers have used non-monetization methods to qualitatively and quantitatively analyze CES preferences and perceptions. Methods include participatory mapping with GIS [13] and the SolVES model [14]. With the efficient economic data mining of big data, CES research methods are becoming more informative [2]. Photos [15] and comments [16] on various platforms, along with historical materials like poems and ancient texts [2], can be used to identify CESs. These data serve as information carriers, recording stakeholders' perspectives on cultural services and landscape features while describing changes over time and space [2,17]. Although photo data are widely used, they cannot capture all CES categories alone. Intangible benefits like inspirational values and spiritual experiences are challenging to identify through visual content alone [18,19]. Natural Language Processing (NLP) analysis can provide more information by explicitly expressing users' motivations, viewpoints, experiences, perceptions, or symbolic meanings [17,20]. Additionally, the NLP method is also commonly used to mine emotions in text and is widely applied on platforms such as Twitter [15] and Instagram [21]. Analyzing visitors' emotions helps understand the demands for CESs in urban ecosystems.

Due to the non-random distribution of CESs in urban ecosystems, they exhibit highly localized patterns, leading to hotspots and cold spots of CES flow [22]. To facilitate the rational allocation of environmental resources within ecosystems, scholars have actively explored the factors that may affect CESs. These features include different land cover types, the size of green spaces, respondents' occupations and ages, and the openness of the landscape [6,23–25]. Notably, various natural and artificial landscapes, such as water bodies, birds, sculptures, and recreational elements, can provide a broad range of CESs, sometimes even more comprehensive than the ecosystem [26,27]. Therefore, by examining visitors' detailed and specific landscape factors and CES preferences, we can better understand the interaction between people and nature in urban parks [26]. Previous studies on CESs and landscape factors in urban parks have primarily focused on a small scale, utilizing offline methods such as face-to-face interviews or participatory mapping [28,29]. These studies often employ manual visual interpretation methods [26,30], which are time-consuming, suitable for small amounts of data, and subject to the inherent subjective biases of researchers [9]. To address this issue, the combination of computer vision and social media photos has begun to be used in extracting perceptual information. However, there is currently relatively little research on the correlation between landscape factors and CESs using this combination [9].

In recent years, scholars have studied how urban park characteristics impact consumer electronics product provision. These studies focus on specific cultural service categories [22,31,32], like aesthetics, or particular urban park types [24,33]. Despite scholars identifying associative rules among CES categories [34,35], there is a lack of comprehensive analysis of the diversity of CES categories and their internal correlations. This may be influenced by data types. Most studies on the correlation between CESs and landscape factors rely on a single data type [9]. Some studies integrate text, tags, and photos [20] but need an analysis process suitable for large-scale data. With social media growth and residents' interest in natural ecology, an efficient process for analyzing large-scale CES perception datasets across time and space is necessary. There were supply and demand differences in CESs in urban parks, but most studies only examined CES supply perception.

For example, Riechers et al. [36] studied the impact of age and residence on the perceived importance of cultural ecosystem services. Research on user satisfaction and expectations (supply) for various CES sub-items is still limited [37]. Considering the aforementioned research gaps, this study focused on urban parks in Hangzhou, attempting a novel research approach—integrating computer vision and text mining methods. We combined the advantages of text and image data to extract CES categories and visitors' emotional attitudes from Weibo comments and landscape factors from Weibo photos to obtain a more comprehensive understanding of CES categories and landscape factors. The research aimed to address the following issues: (1) understanding the CES categories of urban parks as well as the satisfaction and expectations of park visitors; (2) effectively extracting the landscape factor preferences of park visitors; and (3) exploring the impact of internal correlations and landscape heterogeneity within CESs. This study provides a new perspective for evaluating the framework of cultural ecosystems in urban parks, contributing to the coordinated development of public perception and planning management. It offers a scientific basis to rationally allocate environmental resources, optimize the efficiency of cultural services in urban parks, and enhance the well-being of urban residents.

## 2. Materials and Methods

### 2.1. Study Area

Hangzhou (E118°20′–120°37′, N29°11′–30°34′) is located in the Yangtze River Delta, serving as the economic, cultural, and educational center of Zhejiang Province. It is a significant scenic tourist city with rich ecological and cultural resources. Throughout history, the interaction and coexistence of the three major landscape factors (mountains, lakes, and cities) have shaped the long-term relationship between human habitation and the natural environment. The construction of urban parks in Hangzhou has left behind a wealth of perceptual information about the harmonious coexistence of humans and nature. Furthermore, Hangzhou has been a pioneer in constructing urban parks in China. Using Hangzhou as an example, the results of this study have implications that can be valuable for other cities in China and even globally [38].

The study area in this paper was based on the core areas delineated by the 'Hangzhou Green Space System Planning (Revised Edition) (2021–2035)', which include Xihu District, Shangcheng District, Binjiang District, and Gongshu District. These areas are significant carriers of Southern Song culture, Wuyue culture, and Qiantang River culture, with numerous historical sites. These cultural accumulations make these regions rich in tourism resources, with prominent cultural heritage, providing abundant information for CES analysis. In addition, the vegetation structure in the core area is complex, and the vegetation types are diverse. This makes these areas rich in plant resources and high in vegetation coverage, with the West Lake area having the highest forest coverage, reaching 42.60%. Simultaneously, many popular urban parks have been formed during the long process of the transformation and protection of the West Lake Scenic Area in the core area. Therefore, considering three main criteria, we selected representative urban parks within these four districts as our research objects. First, the parks should have a large scale of visitation, with sufficient comment data on social media [19] (filtered for urban parks with posts exceeding 50 pages on Weibo webpages). Second, the parks being studied should have outstanding performance in historical and cultural aspects, plant landscapes, urban characteristics, etc. Third, they should adhere to industry standards. According to industry standards, parks with an area exceeding ten hm$^2$ that can provide various activities are considered comprehensive. Parks with independent land use and an area greater than one hm$^2$ are classified as community parks. Parks with specific themes are categorized as specialized parks. In addition to botanical gardens, zoos, heritage parks, and historic gardens, specialized parks include other specific types such as children's, waterfront, urban forest, and urban wetland parks. Previous academic research has shown that online discussions and visits to community parks are relatively low [39]; hence, they did not appear in this study. Ultimately, based on the screening of the three criteria, we selected twenty-five distinctive

urban parks (Figure 1), categorized into two main groups and eleven subcategories (refer to Appendix A Table A1 for specific classification details).

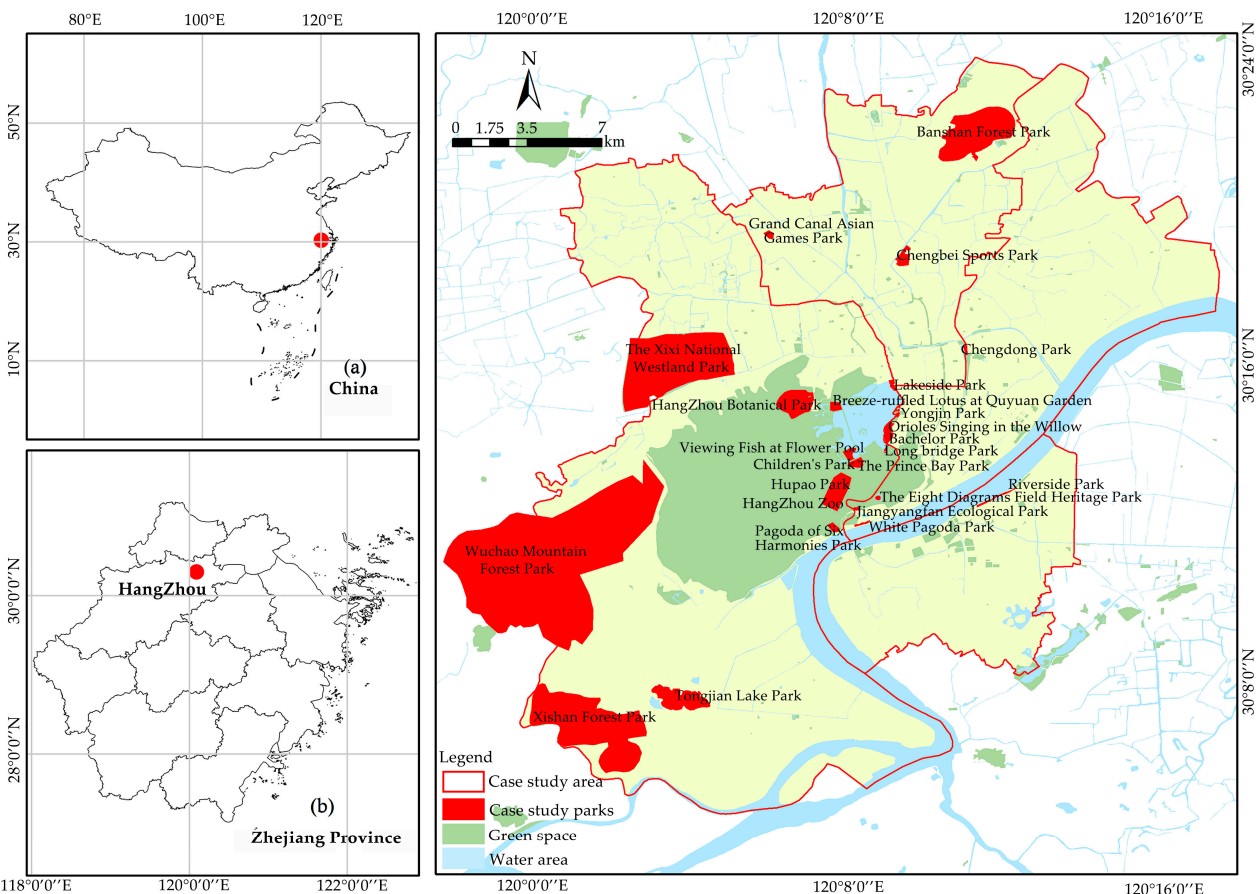

**Figure 1.** Overview map of the research site.

## 2.2. Research Design

This study utilized social media data that combine text and images to correlate CESs with park landscape factors, conducting a perceptual assessment of CESs in urban parks. The research process consisted of the following four steps (Figure 2): (1) Extracting CESs from text using Natural Language Processing (NLP): Jieba 0.42.1 was applied to preprocessed text data to obtain word frequency statistics, and a professional team was employed for coding and constructing a CES perception information lexicon. (2) Using machine learning to identify landscape factors in photos: the YOLO v5 model was trained to extract landscape factors from preprocessed image data and construct a landscape factor perception system. (3) Correlation analysis: SPSS software and IBM SPSS Modeler software were used to conduct an internal correlation analysis of CES categories and a corresponding analysis with landscape factors. (4) IPA evaluation of CESs: using the Baidu AI intelligent platform to conduct a sentiment analysis on text data, visitor satisfaction with CESs was calculated and combined with CES coding statistics (importance) to construct an IPA evaluation structure.

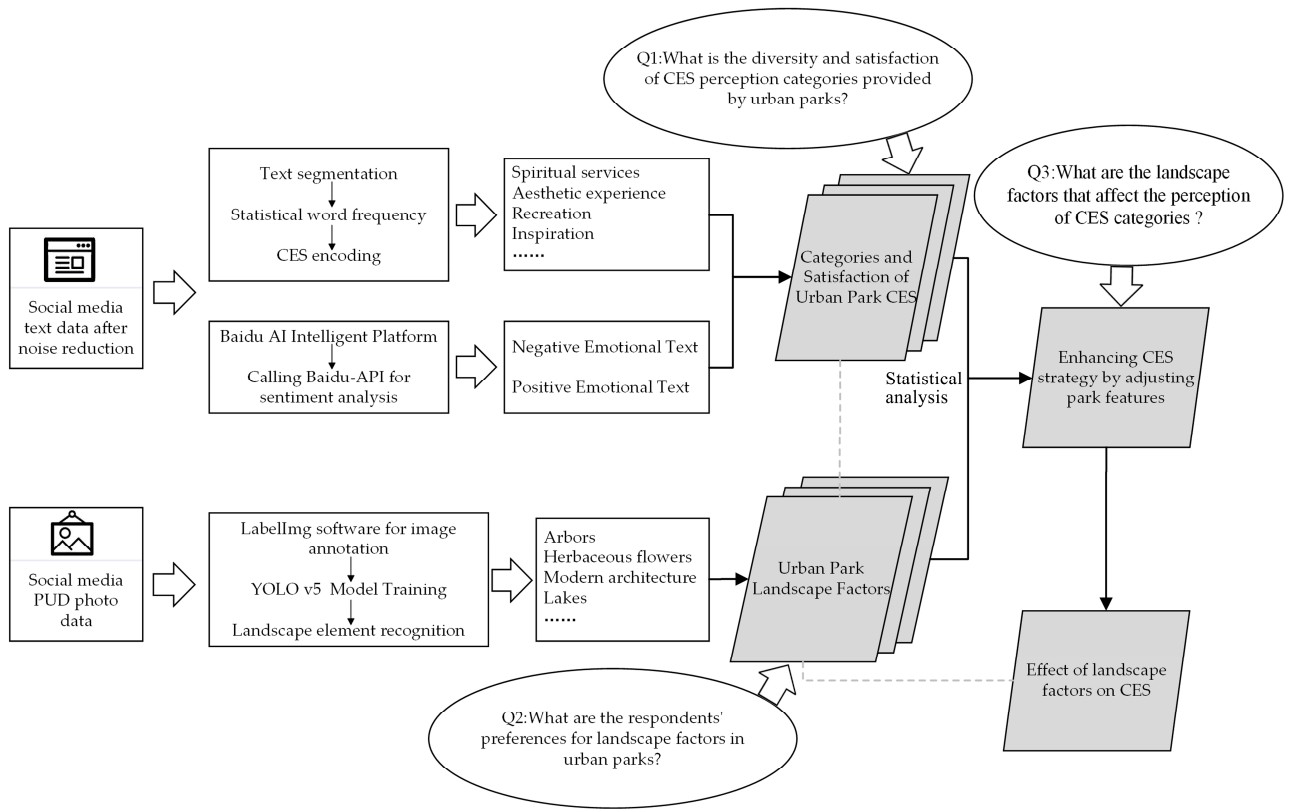

**Figure 2.** Practice-oriented workflow for the case study.

### 2.3. Acquisition of Social Media Data

Social media data, shared by users in digital formats (text, images, and videos), constitute a valuable information source for the collective perception of diverse landscapes [26], serving as a crucial basis for analyzing the perceived cultural services by park visitors. Sina Weibo is the largest social media platform in China, with over 500 million users as of May 2020, accounting for 42.30% of all internet users, providing rich information and data. Therefore, this study used the Python programming language to obtain data from the Sina Weibo application programming interface (API). The data covered a time range from 1 January 2018 to 4 April 2023, with 62,450 metadata items. Raw, crawled social media data often contain a lot of noise and need to be cleaned. We used the Python programming language and Excel statistical software to remove news, advertisements, and data without actual content. At the same time, we also deleted metadata that did not publish photos. Using Panda's library in the program, we traversed the data. Then, we identified and randomly selected User–Day (PUD) photos, i.e., the unique combination of users and dates, to avoid excessive representation of some active users [9]. Finally, we obtained 17,266 analyzable data points. According to the latest Weibo User Development Report released on the official Weibo website, the user demographic on Weibo continued to show a trend toward a younger audience in 2020, with users aged 19–51 accounting for 96.00%. Therefore, this study's Weibo image and comment data primarily originated from middle-aged and young demographics.

### 2.4. Processing of Social Media Data

2.4.1. Framework for Urban Park CESs and Landscape Factor Indicators

We referred to authoritative CES indicator classification schemes such as the Millennium Ecosystem Assessment [5], the International Classification of Ecosystem Services [40], and Ecosystems and Biodiversity Economics [41]. Based on the characteristics of Weibo data and on-site investigations, we finally selected the following nine indicators as the CES categories for this study (Table 1): spiritual services, aesthetic experiences, recreation,

inspiration, existential values, cultural heritage, science education, social interaction, and natural appreciation. Building upon the research of previous scholars on cultural ecosystem services and some landscape factors [42–45], we identified 39 relevant landscape factors. A well-established YOLO model was employed to recognize these 39 landscape factors in photos from 25 parks. Factors such as 'food', 'crowd', 'person', 'sunset', and 'night scene' unrelated to actual park construction were excluded to mitigate the impact of collinear data and outliers. In addition, we also removed the factor of perception frequency below 50 times (considered lower compared to 17,266 photos). Similar factors were merged, for example, combining 'Tree flowers' and 'Color leaf trees' into 'Arbors' and categorizing 'Massifs' as 'Groves'. This process resulted in a final list of 21 landscape factors related to cultural ecosystem services, including lakes, ponds, natural revetments, arbors, herbaceous flowers, and others. Following the classification of landscape design factors by American landscape designer Norman K. Booth [46], these factors were categorized into four categories of urban landscapes: hydrological landscape, biological landscape, architecture landscape, and facility landscape.

**Table 1.** CES perceived information dictionary.

| CES Categories | Indicator Connotation | Code | Partially Matched Keywords |
|---|---|---|---|
| Spiritual services | Forgetting worries and feeling respect for nature. | C1 | Full of vitality, enrich, satisfaction, spirit, leisure, feel grateful |
| Aesthetic experiences | Experience the charming scenery, attractions, sounds or smells here. | C2 | Floral fragrance, beautiful scenery, super good-looking, spring scenery, beautiful |
| Recreation | Conduct various leisure and recreational activities in urban parks. | C3 | Recreation, picnic, take a walk, take a picture, fly a kite, climb mountain, boating |
| Inspiration | The artistic inspiration inspired by the park landscape. | C4 | Inspiration, creation, design, interests, inquisitive, thought, crafty |
| Existential values | A place that can generate a sense of connection and belonging, leading to nostalgia. | C5 | Sense of belonging, landmark building, landmark, cherish the memory of, famous, symbolize |
| Cultural heritage | The cultural and historical value contained in urban parks. | C6 | History, culture, artistic atmosphere, cultural atmosphere, cultural heritage |
| Science education | Obtaining knowledge, science popularization, and educational opportunities through urban parks. | C7 | Education, exhibition, commemoration, study, significance, knowledge, museum |
| Social interaction | Promoting social interaction between people. | C8 | Strolling baby, with family, exchange, share, gather, connection, communicate |
| Natural appreciation | Experience the pleasure of watching or interacting with animals and plants. | C9 | Plant, animal, ornamental flowers, ornamental plant leaves, insect, feed fish, watch fish |

### 2.4.2. Establishment of CES Perception Lexicon

The preprocessed text data underwent secondary processing, removing emoticons and spaces from the text. We used the Jieba 0.42.1 in Python to segment the comment data and annotate parts of the speech. To increase the accuracy of segmentation in the professional domain, based on the relevant literature [17,47,48], we loaded the Harbin Institute of Technology stop word dictionary [49] and a custom dictionary (829 terms) and then replaced synonyms. Due to the extensive database, we conducted word frequency statistics on the segmentation and retained high-frequency words with a frequency greater than six (to avoid ambiguity, the frequency of single-character words was not counted). We referred to the grouping of text word frequency in the literature [9,22,35]. Then, we used the Delphi expert evaluation method [17], established a coding team of 10 professionals,

and coded high-frequency words with a frequency greater than 5 in 25 parks. For disputed parts, we recoded them multiple times until we reached a consensus. Each text could be coded into multiple landscape categories. We used Python programs to convert each text-encoded form and check the CES information of the text that had not yet been encoded. Finally, the CES perception information lexicon was constructed (Table 1).

### 2.4.3. Landscape Factor Extraction Based on Image Annotation

YOLO v5 is currently one of the mainstream target detection models in the YOLO (You Only Look Once) series [50]. Compared to SD and Mobile Net, the YOLO algorithm is a lightweight neural network with few parameters, a fast operation speed, and relatively high accuracy. The YOLO model consists of four parts: the input end, the backbone (primary network), the neck network, and prediction (output end) [51] (Figure 3). In addition to vehicle and pest recognition applications, the YOLO model can effectively identify landscapes. For example, Wang et al. [52] effectively identified images of Mount Lu based on an improved YOLO v4. For this study, we trained the YOLO v5 model using the Anaconda-Py 3.10 environment for Weibo photo data. The target recognition process was as follows: (1) The preparation of a photo training set for landscape factor recognition. We randomly selected 4300 photos from the preprocessed photo data and manually annotated them using Labelimg 1.8.5 annotation software. The landscape labels included 39 categories, such as herbaceous flowers, arbors, lawns, lakes, birds, and identification facilities. These preprocessed photos were randomly divided into training and validation sets in a ratio of 3:1. (2) The dataset was trained for 200 iterations, and a balance was reached at 120 iterations. The accuracy of the validation set converged without significant improvement, indicating that the model training was completed.

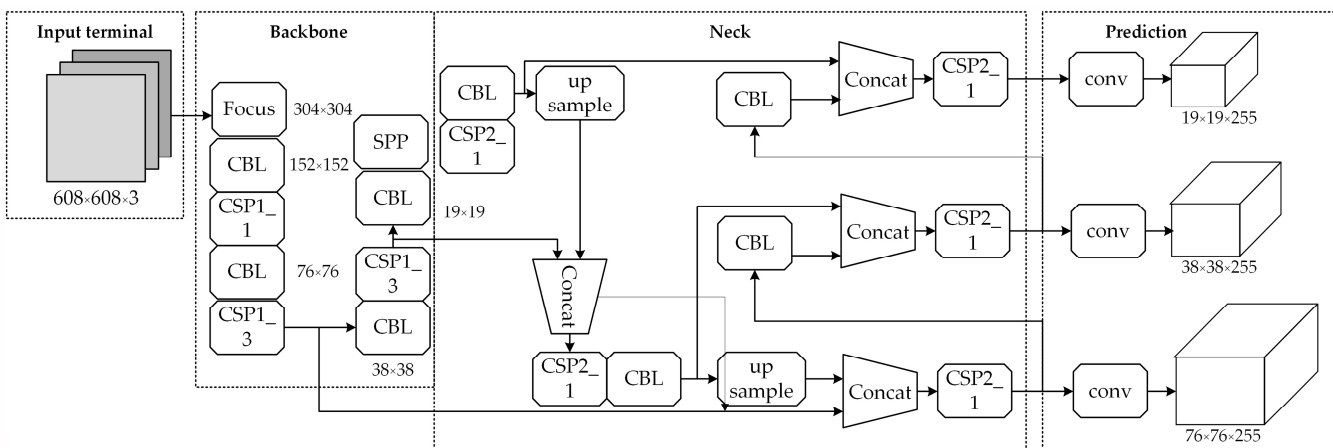

**Figure 3.** YOLO v5 model structure diagram.

We used a confusion matrix to compare the image annotation results performed by the YOLO v5 model with the author's obtained image annotation results to evaluate the accuracy of the image annotation results. Based on the confusion matrix, we calculated True Positives (TP), True Negatives (TN), False Positives (FP), and False Negatives (FN) for the 21 selected landscape factors. TP represent positive results correctly predicted by the model, TN denote negative results correctly predicted by the model, FP represent positive results incorrectly predicted by the model, and FN represent negative results incorrectly predicted by the model. We calculated the precision (Equation (1)), recall (Equation (2)), mAP@0.5, and mAP@[0.5 = 0.95].

$$Precision = \frac{TP}{TP + FP} \tag{1}$$

$$Re\ call = \frac{TP}{TP + FN} \tag{2}$$

*2.5. Data Statistical Analysis*

This study used IBM SPSS Modeler 18.0 software for an Apriori algorithm analysis of internal association rules for CES categories. A Multiple Correspondence Analysis (MCA) could reveal the relationship between multiple categorical variables through the score distances on intuitive graphs. Then, based on the results of the Weibo text data analysis (CES coding statistics, sentiment analysis), we used an IPA to analyze the differences between users' perceptions of the importance of CESs and experience satisfaction. This kind of analysis helps propose targeted strategies for improving the quality of CESs in Hangzhou city parks [53]. Perceived importance was measured by the frequency of occurrences of cultural ecosystem service categories, and experience satisfaction was measured by the proportion of positive text in the text data. The Baidu AI open platform's emotion analysis module completed the emotional judgment of visitors' uploaded texts. All statistical analyses and data plotting were performed in Microsoft Excel 2019 (Microsoft Corporation, Redmond, WA, USA), Python 3.11.4 (Python Software Foundation, Wilmington, NC, USA), SPSS 26 (International Business Machines Corporation, Armonk, NY, USA), and IBM SPSS Modeler 18.0 software (International Business Machines Corporation, Armonk, NY, USA).

## 3. Results

*3.1. CESs in Social Media Texts*

3.1.1. The Similarities and Differences in the Perceptions of Urban Park CESs

The gender ratio of the visitors in the 25 urban parks was 2.26:1, with 57.64% of the visitors being from outside the province and 42.34% from within the province. Out of 17,266 text data points, we counted 7371 words with a frequency greater than six, which were counted and encoded as 985 words as CES perception information. Approximately 14,737 texts reflected visitors' perceptions of CESs, accounting for 85.35%. The overall perception frequency of CESs was above 80.00%, with 13 parks having a perception frequency exceeding 90.00%, accounting for 52.00%. All CES categories were perceived in the various parks, but there were differences among different parks (as shown in Figure 4). The perception frequencies of spiritual services, aesthetic experiences, recreation, and natural appreciation were significantly prominent. Aesthetic experience and recreation had relatively even distribution ratios in the various parks, around 19.00% (Figure 5).

Figure 4 presents the CES frequencies in different types of parks. The comprehensive parks had a total of 7585 text records and 19,412 CES code records documenting visitors' perceptions of CESs. Among them, Prince Bay Park and White Pagoda Park had the highest frequency of perception keywords, accounting for 25.85% and 24.52%, respectively. In the 7152 perception records of specialized parks, there were 21,077 CES code entries, with zoos ($n = 4770$, 22.63%) and botanical gardens ($n = 6431$, 30.51%) showing a significant advantage in the quantity of perceived CES code entries. According to Figure 5, the aesthetic experience function of Tongjian Lake Park and the leisure and recreation function of White Pagoda Park had the highest proportions among all CESs in their respective parks, accounting for 38.16% and 32.04%, respectively. Spiritual services and natural appreciation had average ratios of 12.99% and 14.69%, respectively, ranking only behind aesthetic experience and recreation. The differences in historical and cultural perceptions among the different parks were substantial, with the highest being in the Eight Diagrams Field Heritage Park at 23.96% and the lowest in Hangzhou Zoo at 2.54%. Social interaction was prominent in Children's Park and Hangzhou Zoo, accounting for 17.00% and 12.29%, respectively. These data indicated that the public prefers social activities in these two parks. Among all the parks, only a few visitors perceived inspiration, existential values, or science education (all below 6.20%), with inspiration having the lowest perception frequency.

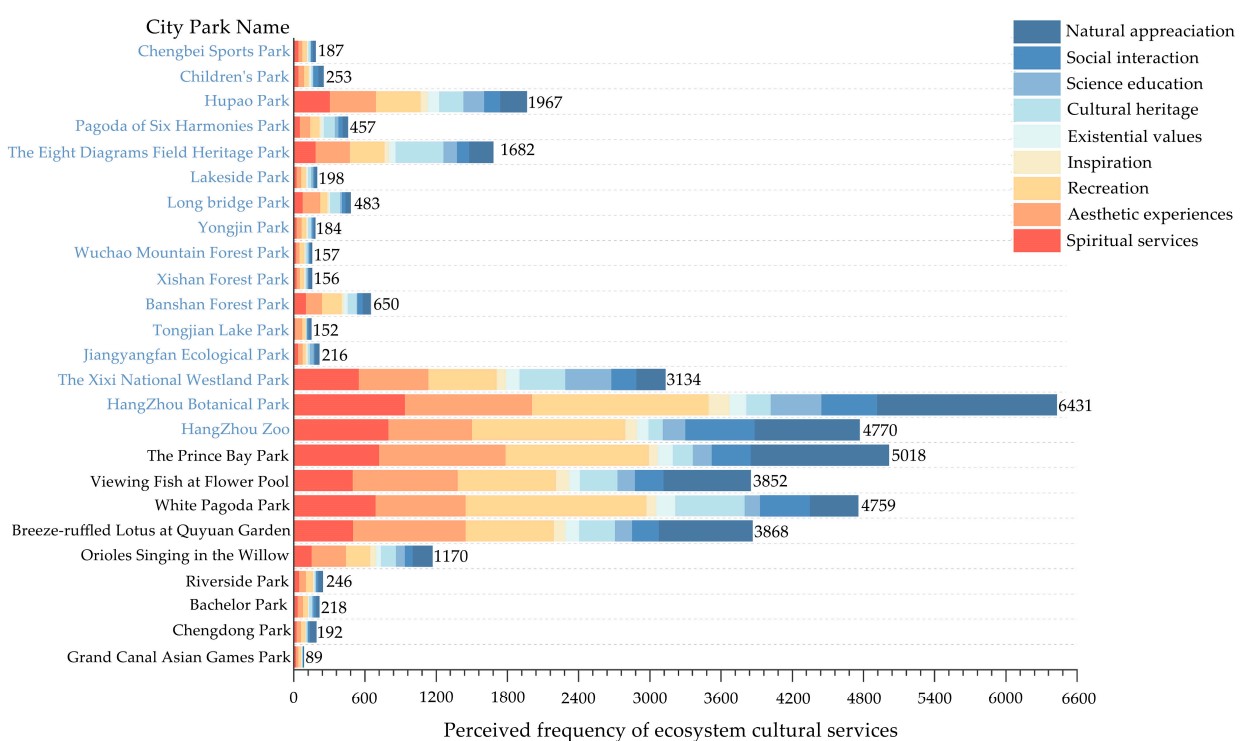

**Figure 4.** Overall perception bar chart of urban park CESs (the park names in blue font are dedicated parks; the park names in black font are comprehensive parks).

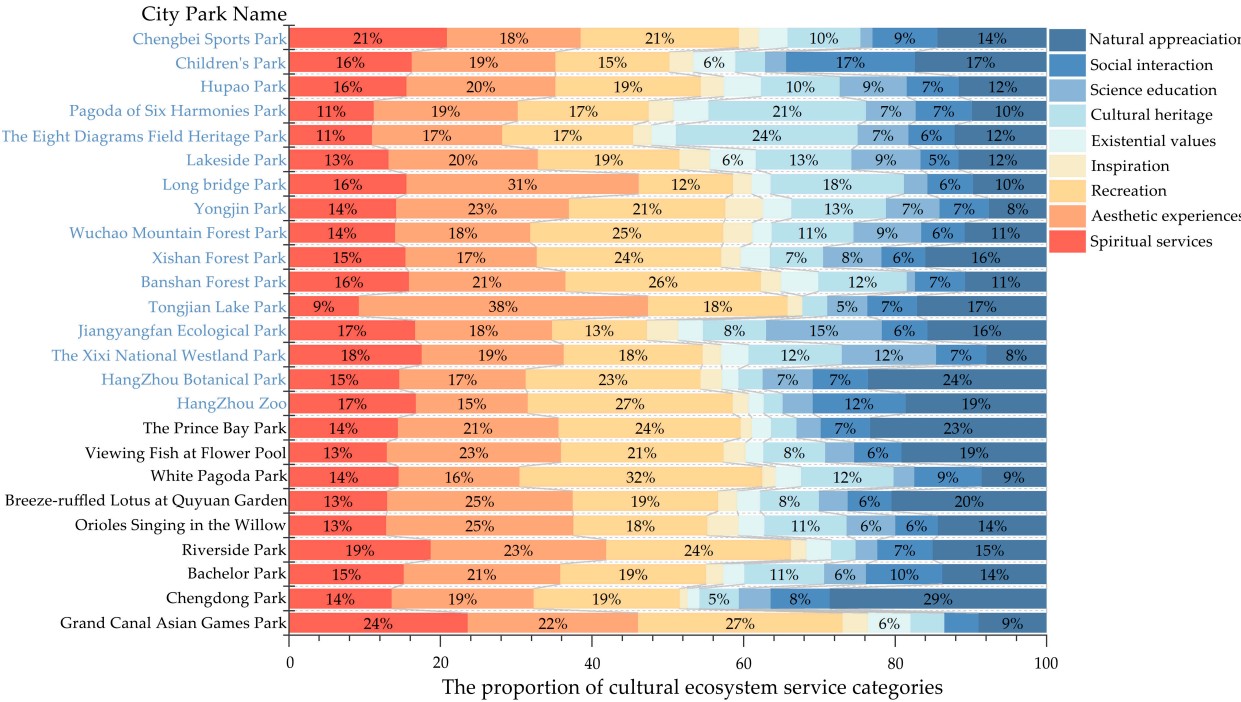

**Figure 5.** Distribution of CES perception types in urban parks (the park names in blue font are dedicated parks; the park names in black font are comprehensive parks).

### 3.1.2. Importance–Performance Analysis of Visitors at Urban Park CESs

According to the emotional value statistics processed by the Baidu AI Open Platform's sentiment analysis module, visitors' positive Weibo texts had the highest proportion (92.07%, *n* = 15,897), followed by negative texts (6.60%, *n* = 1139), and neutral texts (1.32%, *n* = 230). Table 2 shows the results of the visitors' ratings of the importance and perfor-

mance of CESs by type. The overall importance of CESs to all the visitors (24.88) was lower than their performance level (78.97). The satisfaction with recreation and cultural heritage was significantly higher than the other service categories, and the emphasis on aesthetic experiences and recreation was significantly higher than the other service categories. The IPA perception evaluation model was used to analyze the performance of different categories of CESs and the visitors' emphasis. The IPA analysis uses the average values of importance and satisfaction as the dividing points for the X–Y axis, dividing the space into four quadrants [54], corresponding to different management priorities: the first quadrant represents high importance–high satisfaction; the second quadrant represents high importance–low satisfaction; the third quadrant represents low importance–low satisfaction; and the fourth quadrant represents low importance–high satisfaction. According to the perception evaluation model (Figure 6), the CESs provided by urban parks in Hangzhou's main urban area met visitors' needs well. The CES categories that could continue to be maintained include inspiration, existential values, science education, and social interaction; aesthetic experiences and recreation had significant advantages. Service categories that needed improvement included natural appreciation, spiritual services, and cultural heritage.

**Table 2.** Perceived performance and importance of cultural ecosystem services.

| CES Categories | Number of Negative Texts | Number of Active Texts | Importance | Performance (Satisfaction) |
|---|---|---|---|---|
| Spiritual services | 251 | 4868 | 17.37 | 51.00 |
| Aesthetic experiences | 399 | 7474 | 50.00 | 80.67 |
| Recreation | 691 | 9458 | 54.77 | 91.25 |
| Inspiration | 51 | 922 | 5.81 | 80.85 |
| Existential values | 42 | 1120 | 6.81 | 83.42 |
| Cultural heritage | 121 | 3098 | 18.76 | 77.46 |
| Science education | 110 | 2229 | 12.91 | 93.90 |
| Social interaction | 217 | 2807 | 17.79 | 81.74 |
| Natural appreciation | 496 | 7754 | 39.72 | 70.72 |
| Total | 2378 | 3,9730 | 24.88 | 78.97 |

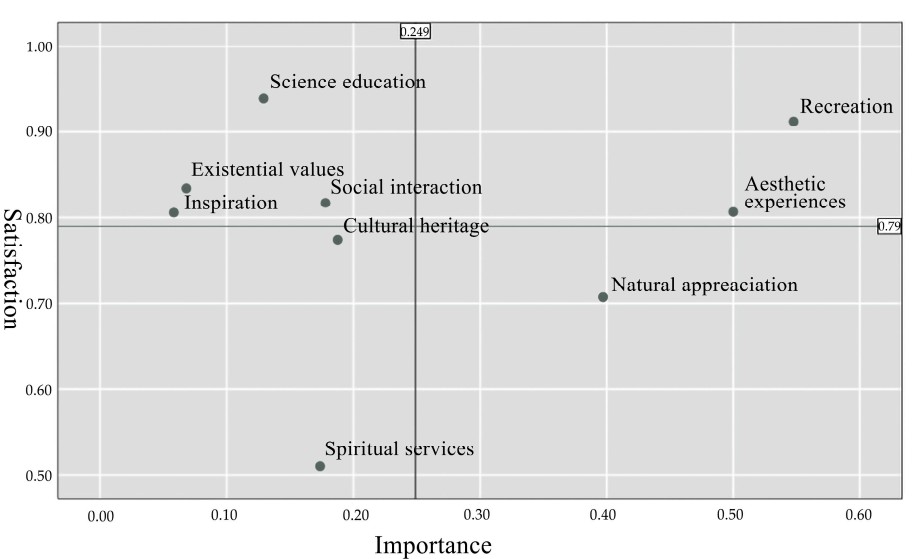

**Figure 6.** The importance and satisfaction of cultural ecosystem service perception.

### 3.2. Landscape Factors in Social Media Photos

After iterative training, the YOLO v5 model had an average precision of 0.981, demonstrating exemplary performance in annotating target images. In verifying 1122 manually annotated images against the model results, all 21 landscape factors showed high precision and recall rates (Table 3). The YOLO v5 model identified landscape factors in 7662 photos out of 14,752 pieces of metadata (excluding data where a CES was not recognized).

Among the identified landscape factors, biological landscape had the highest frequency (61.00%, *n* = 6795), followed by hydrological landscape (16.58%, *n* = 1847) and architectural landscape (15.89%, *n* = 1770). The lowest frequency in the 7662 photos was for facility landscape (6.53%, *n* = 727). As each Weibo post might encode multiple CES categories, we used the Python programming language to match each CES type with landscape factors for each piece of data, resulting in 32,874 corresponding data entries. Among them, biological landscape were associated with the most perceived CESs (*n* = 17,769), while the fewest were found in facility landscape (*n* = 2629) (Table 4). After visualizing Table 4 (Figure 7), we learned that among the four categories of landscapes, hydrological and biological landscape mainly provided perceptions of spiritual services, aesthetic experiences, and recreation. The natural appreciation function was more reflected in biological landscape, while the cultural heritage and recreation functions tended to be reflected in architectural and facility landscape.

**Table 3.** Image annotation results and accuracy.

| Landscape Categories | Landscape Factors | Number of Labels | Accuracy | Recall | mAP@0.5 | mAP@[0.5 = 0.95] |
|---|---|---|---|---|---|---|
| | Lakes | 112 | 0.991 | 0.953 | 0.994 | 0.826 |
| Hydrological landscape | Ponds | 163 | 0.918 | 0.660 | 0.985 | 0.747 |
| | Natural revetments | 263 | 0.890 | 0.889 | 0.952 | 0.626 |
| | Arbors | 438 | 0.961 | 0.947 | 0.985 | 0.790 |
| | Lawns | 69 | 0.985 | 0.927 | 0.983 | 0.769 |
| | Herbaceous flowers | 321 | 0.970 | 0.984 | 0.993 | 0.813 |
| Biological landscape | Woody flowers | 178 | 0.966 | 1.000 | 0.933 | 0.841 |
| | Groves | 99 | 0.980 | 0.878 | 0.988 | 0.730 |
| | Fish | 114 | 0.923 | 0.982 | 0.990 | 0.773 |
| | Birds | 77 | 0.991 | 1.000 | 0.995 | 0.712 |
| | Other animals | 227 | 0.980 | 1.000 | 0.995 | 0.841 |
| | Pavilions | 26 | 0.785 | 1.000 | 0.978 | 0.808 |
| | Bridges | 70 | 0.975 | 0.971 | 0.986 | 0.768 |
| Architecture landscape | Towers | 27 | 0.836 | 0.941 | 0.955 | 0.611 |
| | Other traditional buildings | 220 | 0.949 | 0.986 | 0.993 | 0.778 |
| | Modern architecture | 81 | 0.947 | 1.000 | 0.993 | 0.842 |
| | Entertainment facilities | 127 | 0.965 | 1.000 | 0.994 | 0.811 |
| Facility landscape | Identification facilities | 47 | 0.994 | 1.000 | 0.995 | 0.876 |
| | Seats | 40 | 1.000 | 0.966 | 0.989 | 0.754 |
| | Landscape sketches | 52 | 0.949 | 1.000 | 0.995 | 0.855 |
| | Roads | 110 | 0.956 | 0.982 | 0.994 | 0.789 |

**Table 4.** The occurrence frequency and proportion of CES categories in different landscape categories.

| CES Categories | Hydrological Landscape | | Biological Landscape | | Architecture Landscape | | Facility Landscape | | Total |
|---|---|---|---|---|---|---|---|---|---|
| | N | % | N | % | N | % | N | % | N |
| Spiritual services | 1208 | 3.83 | 3199 | 10.13 | 499 | 1.58 | 318 | 1.01 | 5224 |
| Aesthetic experiences | 1179 | 3.73 | 3455 | 10.94 | 957 | 3.03 | 188 | 0.60 | 5779 |
| Recreation | 984 | 3.12 | 3777 | 11.96 | 799 | 2.53 | 697 | 2.21 | 6257 |
| Inspiration | 180 | 0.57 | 427 | 1.35 | 144 | 0.46 | 129 | 0.41 | 880 |
| Existential values | 146 | 0.46 | 500 | 1.58 | 540 | 1.71 | 150 | 0.48 | 1336 |
| Cultural heritage | 231 | 0.73 | 620 | 1.96 | 1535 | 4.86 | 213 | 0.67 | 2599 |
| Science education | 234 | 0.74 | 575 | 1.82 | 655 | 2.07 | 314 | 0.99 | 1778 |
| Social interaction | 194 | 0.61 | 579 | 1.83 | 368 | 1.17 | 413 | 1.31 | 1554 |
| Natural appreciation | 721 | 2.28 | 4637 | 14.69 | 604 | 1.91 | 207 | 0.66 | 6169 |
| Total | 5077 | 16.08 | 17,769 | 56.27 | 6101 | 19.32 | 2629 | 8.33 | 31,576 |

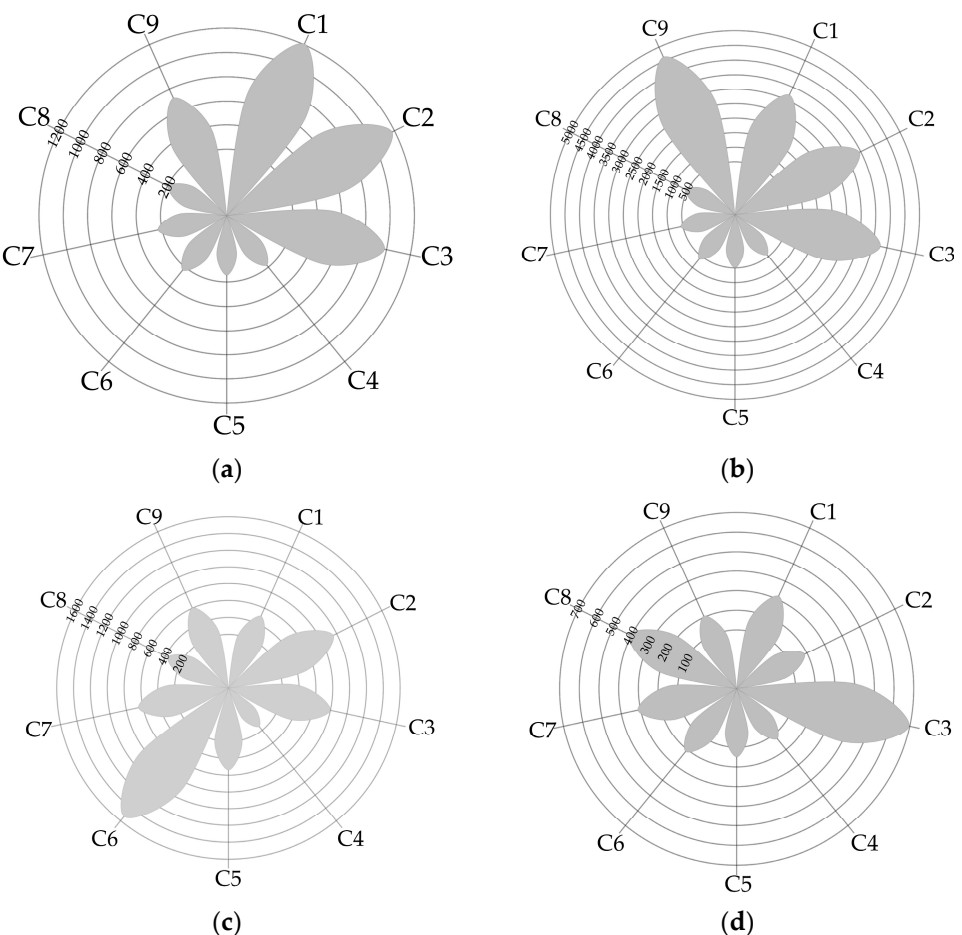

**Figure 7.** Heterogeneity of CES categories under landscape categories. (**a**) Hydrological landscape; (**b**) biological landscape; (**c**) architecture landscape; (**d**) facility landscape; C1: spiritual services; C2: aesthetic experiences; C3: recreation; C4: inspiration; C5: eexistential values; C6: cultural heritage; C7: science education; C8: social interaction; C9: natural appreciation.

*3.3. The Correlation between Urban Park CESs and Landscape Factors*

3.3.1. Interconnectedness of CES Categories in Urban Parks

Through the Apriori association rule analysis of visitors' perceptions of CES categories, 11 association rules that met the conditions (confidence > 60.00% and conditional support > 20.00%) were mined (as shown in Table 5). These 11 association rules indicated strong correlations with a temporal sequence. The most frequent itemsets included aesthetic experiences (C2)–recreation (C3), spiritual services (C1)–aesthetic experiences (C2), and spiritual services (C1)–recreation (C3). This result indicated that in the context of 25 urban parks, these three CES categories were frequently perceived together and had solid correlations and orders. As seen from Table 5, when women perceived aesthetic experiences or spiritual services in urban parks, they were highly likely to perceive recreation. Rules 7–11 reflected the association rules between four categories, and there was a high probability of perceiving another type when perceiving spiritual services with aesthetic experiences or recreation.

We conducted a Spearman correlation analysis to demonstrate the overall correlation among all the CES categories. As seen in Figure 8, there was a correlation among the CES categories, and they exhibited varying degrees of correlation. In addition to the three pairs C1–C2, C1–C3, and C2–C3 with high correlations (*R* > 0.85), the correlation between natural appreciation and aesthetic experiences, recreation, and spiritual services was also high, with *R* values of 0.93, 0.88, and 0.83, respectively. Additionally, the correlation between recreation and social interaction also reached 0.83. However, we found two negative correlations: existential values and inspiration (*R* = −0.07) and inspiration and science

education ($R = 0.00$). This finding suggested that when visitors perceive existential values or scientific education, their perception of inspiration might weaken.

**Table 5.** Mining table for association rules between categories of cultural ecosystem services.

| Number | X | Y | Number of Instances | Support Percentage | Confidence Percentage | Rule Support Percentage | Lift |
|---|---|---|---|---|---|---|---|
| 1 | C2 | C3 | 7825 | 45.32 | 60.79 | 27.55 | 1.12 |
| 2 | C1 | C2 | 5914 | 34.25 | 63.22 | 21.66 | 1.40 |
| 3 | C1 | C3 | 5914 | 34.25 | 68.28 | 23.39 | 1.26 |
| 4 | C2,0 | C3 | 5232 | 30.30 | 61.70 | 18.70 | 1.14 |
| 5 | C1,0 | C2 | 4225 | 24.47 | 61.40 | 15.02 | 1.35 |
| 6 | C1,0 | C3 | 4225 | 24.47 | 68.17 | 16.68 | 1.26 |
| 7 | C9,C2 | C3 | 4055 | 23.49 | 63.87 | 15.00 | 1.18 |
| 8 | C1,C3 | C2 | 4038 | 23.39 | 66.86 | 15.64 | 1.48 |
| 9 | C9,C3 | C2 | 3874 | 22.44 | 66.86 | 15.00 | 1.48 |
| 10 | C1,C2 | C9 | 3739 | 21.66 | 60.76 | 13.16 | 1.53 |
| 11 | C1,C2 | C3 | 3739 | 21.66 | 72.21 | 15.64 | 1.34 |

Note: 0 is female; C1 is spiritual services; C2 is aesthetic experiences; C3 is recreation; C9 is natural appreciation.

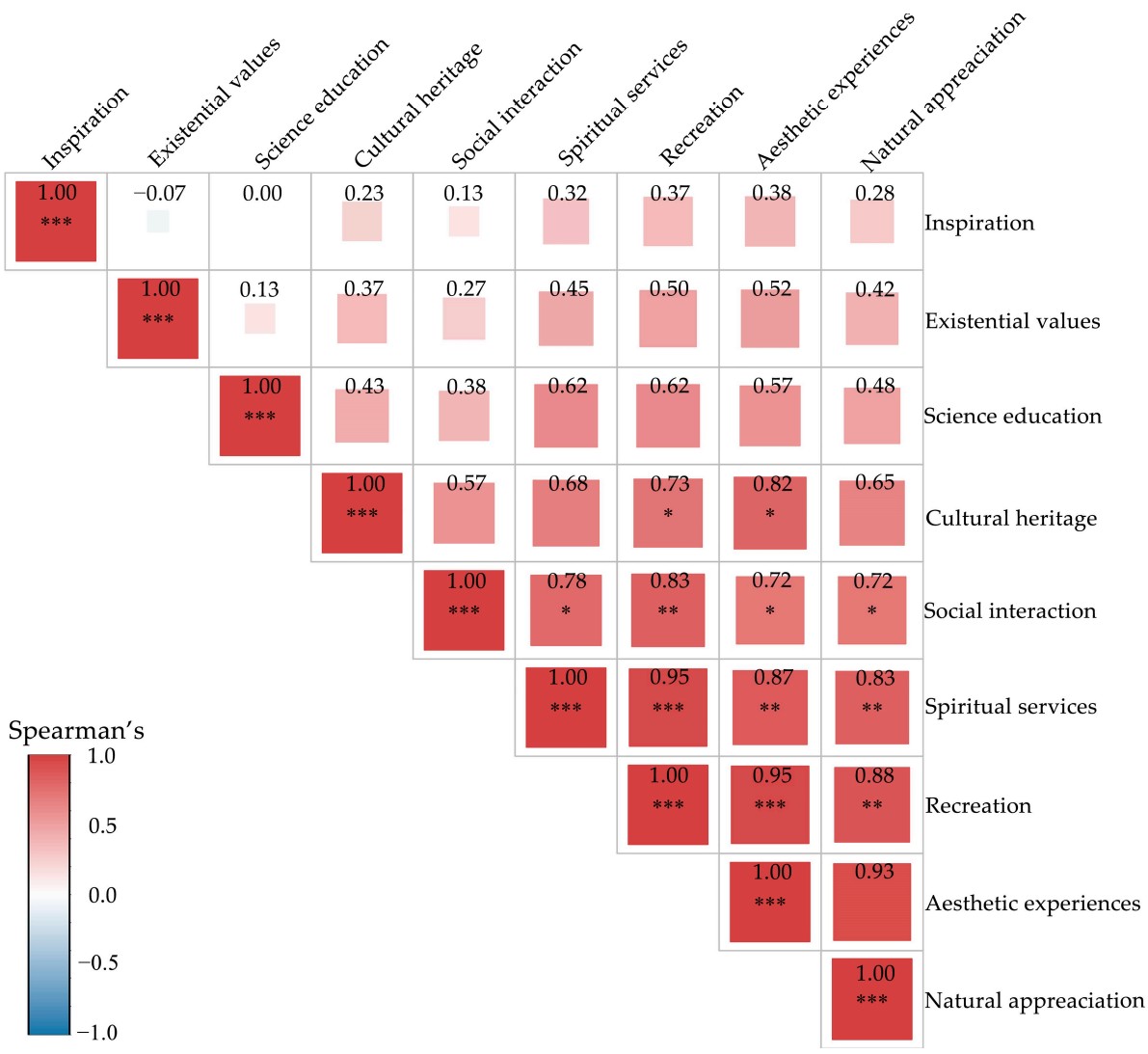

**Figure 8.** Correlation between categories of cultural ecosystem services. Note: ***, $p < 0.001$; **, $p < 0.01$; *, $p < 0.05$.

### 3.3.2. Correlation between CES Categories and Park Landscape Factors

Cross-tabulation and Monte Carlo post hoc chi-square tests showed a significant statistical correlation between the CES categories and landscape factors ($\chi^2 = 21{,}156.983$, df = 160, $p < 0.05$). We used an MCA analysis and the Euclidean distance measurement method to associate landscape factors in photos with CES categories in text data. The results of the analysis showed that three components account for 39.11% of the total variance in visitors' expression of perceived CES categories (Figure 9 shows the dual plots of the first two axes). The first part of Figure 9 (accounting for 52.39% of the total variance) revealed gender differences in visitor perception. Component 2 (accounting for 48.58% of the total variance) showed the perceived connection between landscape factors and cultural ecosystem service categories.

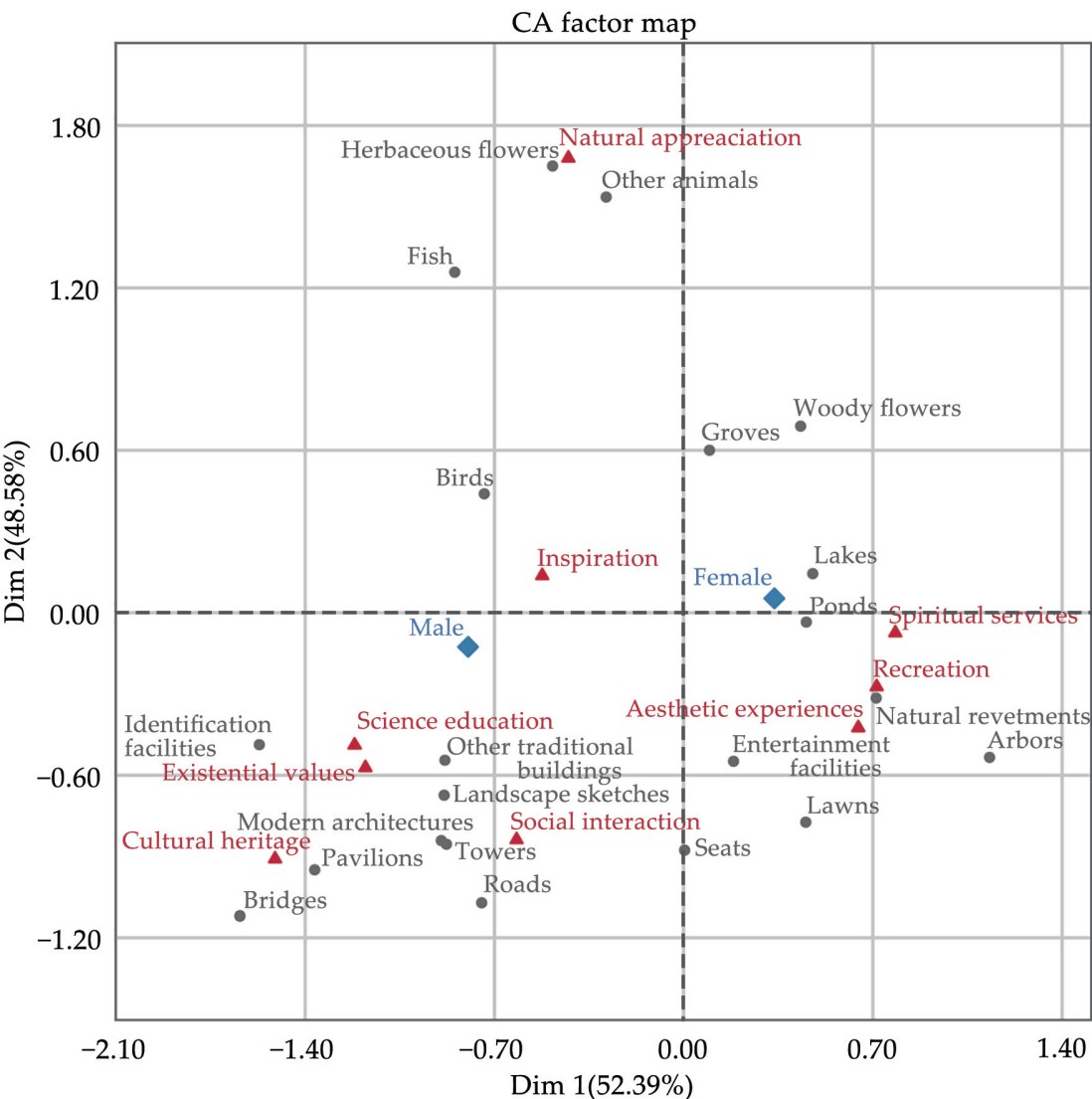

**Figure 9.** The MCA diagram shows the positions of CESs (red font), landscape factors (gray font), and visitor gender (blue font) in the first two-dimensional factor plan.

The MCA indicated that user gender, landscape factors, and perceived CES categories were closely related. Concerning gender landscape preferences, males and females were in the third and first quadrants, respectively, with some differences. Males preferred architectural landscapes (other traditional buildings, modern buildings, pavilions, bridges, towers), landscape sketches, identification facilities, and roads. At the same time, females were more inclined to perceive hydrological landscapes (lakes, ponds, natural revetments),

groves, and woody flowers. Regarding CES categories, males had a higher perception of science education, existential values, cultural heritage, and inspiration, while females were likelier to perceive recreation, aesthetic experiences, and spiritual services. In studying the correlation between CESs and landscape factor perception, we could judge the degree of correlation based on the distance to the origin [55]. From the MCA dual-line chart (Figure 9), we found that natural appreciation was closely related to herbaceous flowers, other animals, and fish, showing a significant correlation. Recreation, spiritual services, and aesthetic experiences form a bundle closely related to ponds, natural revetments, lawns, and groves. Additionally, science education, existential values, and cultural heritage form another bundle associated with identification facilities, landscape sketches, and architectural landscapes (other traditional buildings, modern buildings, pavilions, towers, and bridges). Pavilions and bridges were closely connected to historical and cultural factors, while towers, modern architecture, and roads were closely linked to social interaction. Traditional buildings and signage facilities were closely associated with science education and existential values.

The author created a Sankey diagram to visually represent each CES category's perceptual flow. As shown in Figure 10, females contributed significantly to the perception of all the CES categories, enhancing the perceptions of spiritual services, aesthetic experiences, and recreation for all landscape factors, particularly arbors and herbaceous flowers, fish, landscape sketches, and seats, which had the lowest contribution rates to the perceptions of the remaining six CES categories.

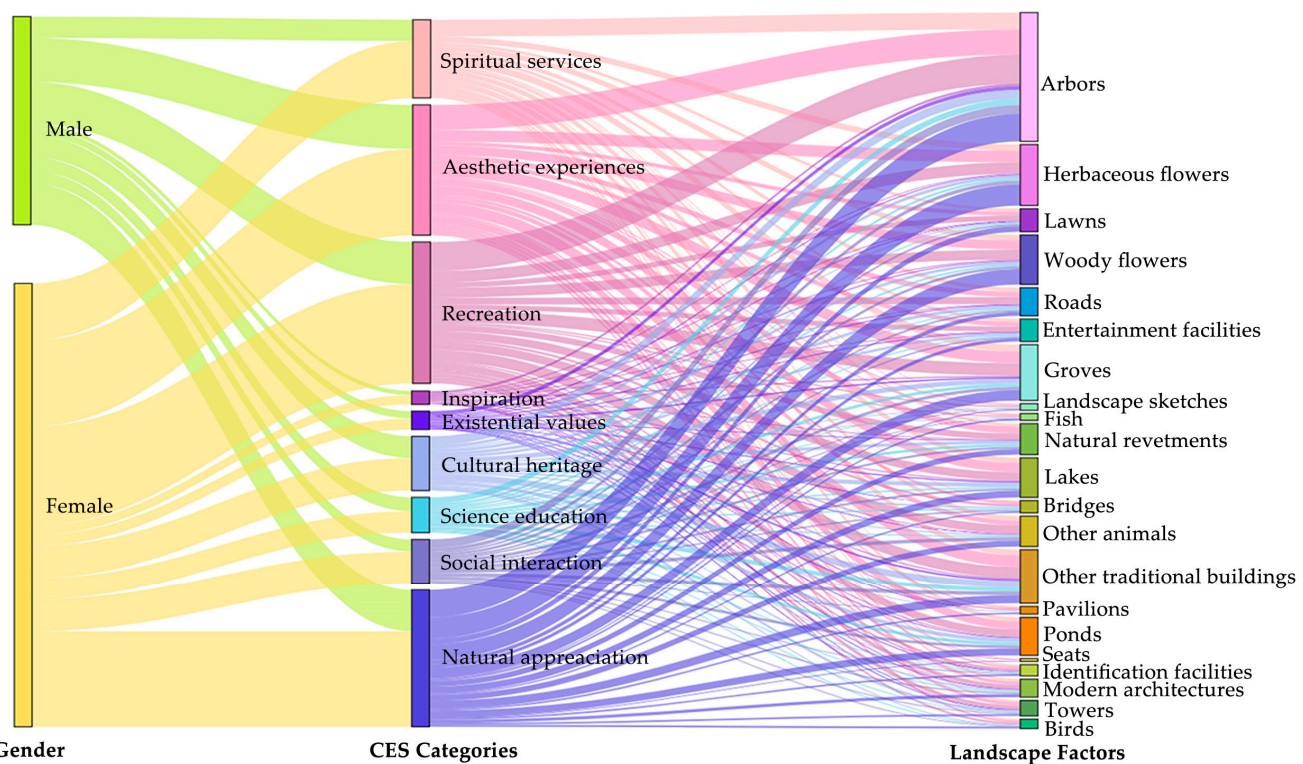

**Figure 10.** Cultural ecosystem services and landscape factors, gender relationship flow.

## 4. Discussion

### 4.1. Visitors' Preferences for Cultural Ecosystem Service Categories and Landscape Factors in Parks

In previously published studies, visitors' perceptions of CESs, entertainment, and aesthetics have often been highlighted, while capturing inspiration was less common [24,56]. These research results partially aligned with our findings. Through using NLP methods to identify comments, we found that the overall perception frequency of Hangzhou urban parks' CESs was relatively high. Among them, spiritual services, aesthetic experiences, recreation, and natural appreciation were most easily perceived by the public. At the

same time, inspiration and existential values were relatively challenging to perceive. This phenomenon was related to Hangzhou being one of China's famous historical and cultural cities and tourist destinations. Yao and Leng et al. [57] found that parks with high-level management, service environments, and green coverage could bring residents pleasant mental experiences. In contrast to the results of Gai et al. [3], this research found that visitors' overall perception of the importance of CESs was lower than their actual performance. This result was related to the high-quality construction and service management of urban parks in Hangzhou. According to the IPA analysis, visitors had higher expectations for natural appreciation, spiritual services, and cultural heritage, but the actual performance of parks in these aspects was slightly lower. This phenomenon might be attributed to factors such as a high prevalence of mosquitoes in the plant community and a lack of systematic introduction and interpretation of historic and cultural landscapes in the parks. In future urban park construction and management, in addition to introducing mosquito-repelling plants in close-range viewing areas, using innovative educational methods such as WeChat accounts for dissemination and QR code technology to share information about flora and historical buildings can also enhance visitor satisfaction with natural appreciation, science education, and cultural heritage.

According to the statistics of 17,266 user–day (PUD) photos, three landscape factors, including arbors (19.01%), herbaceous flowers (8.99%), and groves (8.22%), were more frequently depicted in landscape factors. This was consistent with the findings of Oteros-Rozas et al. [26], Ma et al. [45], and Graves et al. [58], who found that the public highly perceived plant landscapes such as trees. The degree of preference for different landscape factors was similar to the conclusion of Huang et al. [59], who found that the preferred landscape factors in urban parks were deciduous trees > herbaceous flowers > ponds > gates > pavilions. Daniel et al. [60] found that over 20.00% of photos were taken of nature, and natural photos were more likely to occur in parks and areas with high vegetation cover. Similar to this discovery, we also extracted a relatively high proportion of natural landscapes from the photo data. At the same time, unlike the conclusion of Yang et al. [61], there were some differences in landscape category preferences based on visitor gender. Males preferred architectural landscapes, facility landscapes, and roads. At the same time, females were more inclined to perceive hydrological landscapes, groves, and woody flowers. This difference might be related to gender habits. Yang et al. [62] found that most male tourists preferred sitting, while female tourists preferred taking photographs. In Hangzhou, urban parks, flower clusters, and natural embankments often attract visitors to pause and take pictures. Urban park managers can capitalize on gender-specific landscape preferences by strategically placing service facilities in different recreational areas. For instance, they can create prominent photo spots in flower landscapes favored by women and install nearby restroom facilities to enhance the visitor experience.

*4.2. Factors Influencing Visitors' Perception of Cultural Ecosystem Services in Park Landscapes*

The research indicated that a set of ecosystem service functions can consistently co-occur in time and space and could be perceived concurrently. This aligned with conclusions from studies by Jang et al. [35], Zhai et al. [63], and Wang et al. [64]. The Apriori association rule analysis revealed a significant correlation between aesthetic experiences, recreation, and spiritual services. This result resembled the conclusion that nature-based tourism affects visitors' happiness [65]. Unlike Fulvia et al. [34], a strong association was found between natural appreciation, aesthetic experiences, and recreation. When users perceived natural appreciation and recreation, there was a high probability of perceiving aesthetic experiences. This might be due to the introduction of flowering plants or trees species in urban parks, which enhanced park attractiveness and conveyed aesthetic experiences [22]. Previous research on demographic characteristics and CESs found gender to be a crucial factor influencing public perception [66]. This study found females were more likely to perceive CESs than males, contrasting with Tan et al. [67] but aligning with Calvet-Mir et al. [68]. Exploring the preference differences in CES categories, females preferred



aesthetic experiences, recreation, and spiritual services, while males preferred cultural heritage, science education, existential values, and social interaction. Researchers drew divergent conclusions about gender preferences [33,69], which might be influenced by age, occupation, cultural differences, etc.

In this study, an MCA analysis was used to bind CES categories with landscape factors, revealing a close relationship between landscape factors and CES category perception. Previous research [23,43] found that all 21 landscape factors influenced CES category perception to varying degrees, and the same landscape factor could influence multiple CES categories. Among them, the main factors influencing CESs were arbors, herbaceous flowers, woody flowers, other traditional buildings, natural revetments, towers, landscape sketches, bridges, and other animals. Numerous studies have demonstrated these landscape factors to be generally preferred by the public [45,70,71]. Overall, associations formed three bundles of CES and landscape factors. As expected, the perception of natural appreciation mainly came from natural landscapes (biological and water). These landscape factors often coexist because environments such as lakes and forests are habitats for various animals. Similar to the findings of Zhang et al. [23], natural areas did not have a significant advantage in contributing to their existential values compared to non-natural areas such as landscape sketches, buildings, roads, and identification facilities. Science education, cultural heritage, and social interaction often trigger perceptions related to human-made landscapes [26]. Regrettably, we found that visitors' perception of science education was mainly related to artificial landscapes, such as identification facilities, and they rarely paid attention to the knowledge provided by natural factors, such as plants in urban parks. The perception of cultural heritage came mainly from landscape factors such as identification facilities and buildings (pavilions, bridges, towers, and modern buildings). These landscapes were usually carriers of poetry and scientific texts. Some identification facilities recorded information about plants and structures, while certain buildings were endowed with personified historical images and spiritual representations [28]. Visitors intuitively perceived these landscape factors as historical and cultural, and they were also considered to have good educational and scientific uses [72]. At urban life's brisk pace, few tourists felt the urge to explore historical knowledge, except for professionals. Urban park staff can strengthen science education by promoting experiential learning through practice, such as knowledge tests and architectural pattern design competitions, to enhance their understanding of architecture and biological landscapes through participatory activities. In the third bundling group, landscapes contributing to recreation, aesthetic experiences, and mental satisfaction shared similarities. These similarities primarily came from hydrological features and vegetation, including natural revetments, ponds, lawns, and arbors. Similar studies have also extensively reported on the importance of vegetation and water bodies for aesthetics [73,74]. As suggested by Zhang et al. [75], planting more trees and vibrant flowers and providing transparent water sources could enhance public aesthetics and improve mental restoration.

### 4.3. Methodological Advantages and Limitations

In recent years, social media data have become a new trend in researching CESs. Compared to traditional data, social media text data have the advantages of a large volume, long span, accessible collection, and strong randomness [17] and have gradually become an important data source for studying CES perception. This study combined text data with photo data and used machine learning to advance more accurate photo content recognition. This process overcame the limitation that a single type of data might cause the public to lose some perceptual information. Li et al. [76] focused on analyzing the text and image data of Weibo and proved that these two types of data can comprehensively describe public aesthetic elements. Landscape studies have indicated that this method could effectively reflect users' perceptions of CESs and landscape factors [20]. Furthermore, by assessing CES and landscape factor perceptions, we can comprehensively understand

human interactions in urban parks [3], providing decision makers with more targeted transformation suggestions and a theoretical basis.

While this study proposed a method for combined text and image CES research, it had limitations related to online data. Uploaded data content is influenced by visitor characteristics like education level, gender, income, and visit frequency. Obtaining such data is challenging on privacy-focused social media platforms, reflecting the current state of most platforms [17,24]. Social media data mainly reflect the opinions of younger users who publish comments, making it challenging to study the perceptions of users like children and the elderly who do not often comment. Future research could employ various methods, including surveys, participatory mapping, and social media evaluation for different user groups. Integrating multiple data sources, including mobile signaling and exercise app data [17], can enhance CES perception evaluation methods. Refining factors influencing public CES perception will provide park managers with practical suggestions for improving CES supply and spatial quality in urban parks.

## 5. Conclusions

Cultural ecosystem services do not represent the quality of the ecological environment but rather reflect the complex and dynamic interactions between ecosystems and humans. This study analyzed park visitors' preferences for CESs provided by urban parks and the impact of landscapes on these preferences from a first-person perspective. We found that utilizing two types of data from the Weibo platform to explore visitors' perceptions of CESs and landscape factors was an effective method. In addition to different association rules within CES categories, the perception of CES categorieswas also closely correlated with visitor gender and landscape factors. This correlation led to bundling multiple landscape factors with several CES categories. To improve the quality of park ecological services and meet the needs of residents, the construction and management of urban parks should carefully consider the preferences of park visitors towards CES and landscapes. Detailed landscape factors can be employed to target improvements in specific CES categories. However, due to the limitations of the social media user population, the research cannot deeply understand the influence of personal information factors. In subsequent steps, a combination of participatory mapping and various data types, such as mobile signaling, should be employed for a broader analysis to enhance the practical significance of the research results.

**Author Contributions:** Conceptualization, Y.C., Y.Z. and F.S.; methodology, Y.C.; software, Y.C.; validation, Y.C. and Y.Y.; formal analysis, Y.C.; investigation, Y.C., C.H., Y.Y., J.L., Y.W. and T.Z.; resources, Y.C. and C.H.; data curation, Y.C. and C.H.; writing—original draft preparation, Y.C.; writing—review and editing, Y.C.; visualization, Y.C., C.H. and Y.Y.; supervision, Y.Z. and F.S.; project administration, Y.Z. and F.S.; funding acquisition, F.S. All authors have read and agreed to the published version of the manuscript.

**Funding:** This research was funded by the National Key R&D Program of China, grant number: 2022YFF1303102.

**Data Availability Statement:** The data presented in this study are available on request from the corresponding author. The data are not publicly available due to privacy reasons.

**Acknowledgments:** We thank Cong Zhang and Juan Yao for their valuable comments and suggestions on the manuscript. We also express our gratitude to Shanghai Tengyun Biotechnology Co., Ltd. for developing the Hiplot Pro platform and providing technical assistance and valuable tools for the data analysis and visualization.

**Conflicts of Interest:** The authors declare no conflicts of interest.

## Appendix A

**Table A1.** Detailed information on 25 city parks.

| District | Urban Park Name | Area/hm² | Types | Number of Valid Comments |
|---|---|---|---|---|
| Xihu District | The Prince Bay Park | 80.00 | Comprehensive park | 2497 |
| | Viewing Fish at Flower Pool Park | 20.00 | Comprehensive park | 1548 |
| | Breeze-ruffled Lotus at Quyuan Garden | 12.65 | Comprehensive park | 1901 |
| | Hupao Park | 13.12 | Other specialized park (theme of Tiger Running Spring) | 619 |
| | HangZhou Zoo | 20.00 | Specialized park (zoo) | 2195 |
| | HangZhou Botanical Park | 284.64 | Specialized park (botanical park) | 2991 |
| | Youth & Children's Park | 14.00 | Other specialized park (children's park) | 81 |
| | Agoda of Six Harmonies Park | 6.80 | Specialized park (heritage park) | 108 |
| | Xishan Forest Park | 1381.00 | Other specialized park (urban forest park) | 52 |
| | Wuchao Mountain Forest Park | 522.00 | Other specialized park (urban forest park) | 47 |
| | Tongjian Lake Park | 67.72 | Other specialized park (urban wetland park) | 59 |
| | The Xixi National Westland Park | 1150 | Other specialized park (urban wetland park) | 1026 |
| Shangcheng District | White Pagoda Park | 78.40 | Comprehensive park | 2246 |
| | Chengdong Park | 6.26 | Comprehensive park | 69 |
| | Orioles Singing in the Willow | 21.00 | Comprehensive park | 510 |
| | Bachelor Park | 12.70 | Comprehensive park | 68 |
| | Chengbei Sports Park | 44.78 | Other specialized park (sports and gymnastic park) | 52 |
| | Jiangyangfan Ecological Park | 19.80 | Other specialized park (urban wetland park) | 56 |
| | Long bridge Park | 2.80 | Other specialized park (waterfront park) | 203 |
| | The Eight Diagrams Field Heritage Park | 6.00 | Specialized park (heritage park) | 435 |
| | Lakeside Park | 5.80 | Other specialized park (waterfront park) | 53 |
| | Yongjin Park | 2.80 | Other specialized park (waterfront park) | 53 |
| Gongshu District | Grand Canal Asian Games Park | 46.70 | Comprehensive park | 33 |
| | Banshan Forest Park | 1002.88 | Other specialized park (urban forest park) | 257 |
| Binjiang District | Riverside Park | 28.00 | Comprehensive park | 107 |

Although the area of Chengdong Park does not exceed 10 hm², it is stated in the government announcement that it is a comprehensive park. (https://wgly.hangzhou.gov.cn/art/2022/1/17/art_1229505585_58939557.html, accessed on 12 December 2022).

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
