# Peer review of "Mining Social Media Data to Capture Urban Park Visitors’ Perception of Cultural Ecosystem Services and Landscape Factors"

_forests, doi:10.3390/f15010213_

Round 1

Reviewer 1 Report

Comments and Suggestions for Authors

Greetings,

The paper is well written. All selections are well written and well explained. The abstract contains all the elements and the paper is well explained. In the introduction, it is necessary to emphasize the aim of the research and the contribution of the research in order to introduce the readers to the topic of this paper. Explain first why this area was taken into consideration in one papus and then explain. After that, it is still necessary to state in the conclusion what the limits were in this research and to give guidelines for future research.

All the best.

Author Response

Dear Editors and Reviewers,

  We greatly appreciate your critical review of our manuscript (Forests-2785507). Thank you for your helpful comments and constructive suggestions, which helped improve the English of the manuscript and the quality of the paper.

  We have revised the manuscript according to the reviewers’ comments. The attached file includes all the figures in the manuscript, a copy of the revised version with the changes marked and and a response to the reviewer's comments.

  With all the revisions made, we are hopeful that the manuscript will be acceptable for publishing. Thank you for your guidance throughout the revision process.

Sincerely,

Feng Shao

Email address: [email protected]

Reviewer 2 Report

Comments and Suggestions for Authors

The author uses Weibo-based visual media data to develop cultural ecosystem services in Hangzhou city parks. Overall, the author's paper has a reasonable design, reliable results, and has certain research significance for urban park planning and design. My main questions are:

(1) In the introduction, the author still does not point out the research gaps in the research topic. This is also the focus of all readers.

(2) The basis for selecting the 25 urban parks needs further explanation? When it comes to Hangzhou city parks, West Lake is definitely the most famous. Why was it not chosen?

(3) Figure 4 is recommended to be redrawn as it is unclear and has poor readability. The size of each sub-figure in Figure 7 remains consistent.

(4) The discussion recommends increasing comparison with previous results, and how should the cultural ecosystem services of urban parks be improved in a targeted manner?

(5) In the conclusion, it is recommended to delete the future outlook and move it to the discussion. And further reduce the content of the conclusion.

Author Response

(The authors gave the same response as above.)

Reviewer 3 Report

Comments and Suggestions for Authors

Urban parks have a variety of ecosystem services, especially Cultural Ecosystem Services(CES), which are important for the physical and mental health of residents. Based on data from 25 urban parks in Hangzhou, this article reports an original study of the public perceptions of landscape factors and CES. Combining computer vision with text mining, this article assess the preferences and correlations between visitor-perceived CES and park landscape factors, so i feel this is an interesting research. However, the manuscript needs major revision before it would be suitable for publication.

 (1) In the introduction section, it is suggested that the summary and generalization of existing relevant studies should be strengthened. In fact, there have been many similar studies, and it is necessary to introduce the advantages and characteristics of this article.

(2) I think it is necessary to introduce the basis for the selection of parks, such as how many reviews of the park should be considered. At the same time, parks can be classified to express the representativeness of these parks.

(3) In the materials and methods section, please describe in detail the landscape classification framework and process of urban parks.

(4) There is a phenomenon that needs to be explained, and teenagers and the elderly in general do not participate in the comments, so it is necessary to present the representation of the residents in the article.

(5) In the discussion section, it is recommended to explore the reasons why landscapes are more likely to be perceived and preferred. Meanwhile, some landscape management strategies can be proposed.

       (6)Moderate English changes required.

Comments on the Quality of English Language

Moderate English changes required.

Author Response

(The authors gave the same response as above.)

Reviewer 4 Report

Comments and Suggestions for Authors

The subject is relevant for the journal as it approaches green infrastructure and urban parks which may contain urban forests.

The abstract summarises well the paper.

The topic is very original. The concept of "cultural ecosystem services" is not very spread, still the authors found valuable references to it. It is desirable that in the revision the connection between "cultural ecosystem services" and "cultural landscape" with all charters connected to it to be highlighted.

Another item which deserves more in depth view is the connection to well-being. The contribution of landscape architecture to well-being has been approached from various angles, including its role in the pandemic, and it is desirable to narrow down which aspects of well-being this analysis refers to.

Other than that the paper is well structured, the methods are solid and well described, the criteria are well set and followed.

Author Response

(The authors gave the same response as above.)

Round 2

Reviewer 3 Report

Comments and Suggestions for Authors

Please check the English grammar and increase the image resolution.

Author Response

Dear Editors and Reviewers,

  We sincerely appreciate your valuable feedback on our manuscript (Forests-2785507). Thank you for your constructive comments and suggestions, which have been instrumental in improving the quality of the manuscript.

  We have made revisions to the manuscript based on the reviewer's comments. The accompanying files include all the figures from the manuscript, a revised version with tracked changes, and responses to the reviewer's comments.

[email protected]
